# Rapid and Non-Invasive Detection of *Aedes aegypti* Co-Infected with Zika and Dengue Viruses Using Near Infrared Spectroscopy

**DOI:** 10.3390/v15010011

**Published:** 2022-12-20

**Authors:** Gabriela A. Garcia, Anton R. Lord, Lilha M. B. Santos, Tharanga N. Kariyawasam, Mariana R. David, Dinair Couto-Lima, Aline Tátila-Ferreira, Márcio G. Pavan, Maggy T. Sikulu-Lord, Rafael Maciel-de-Freitas

**Affiliations:** 1Laboratório de Mosquitos Transmissores de Hematozoários, Oswaldo Cruz Institute, Oswaldo Cruz Foundation, Rio de Janeiro 21040-360, Rio de Janeiro, Brazil; 2School of Biological Sciences, University of Queensland, Brisbane, QLD 4072, Australia; 3Spectroscopy and Data Consultants Pty Ltd., Brisbane, QLD 4035, Australia; 4Department of Arbovirology, Bernhard Nocht Institute of Tropical Medicine, 20359 Hamburg, Germany

**Keywords:** *Aedes aegypti*, Near Infrared Spectroscopy (NIRS), dengue, Zika, chikungunya, early-warning system, surveillance, diagnostic

## Abstract

The transmission of dengue (DENV) and Zika (ZIKV) has been continuously increasing worldwide. An efficient arbovirus surveillance system is critical to designing early-warning systems to increase preparedness of future outbreaks in endemic countries. The Near Infrared Spectroscopy (NIRS) is a promising high throughput technique to detect arbovirus infection in *Ae. aegypti* with remarkable advantages such as cost and time effectiveness, reagent-free, and non-invasive nature over existing molecular tools for similar purposes, enabling timely decision making through rapid detection of potential disease. Our aim was to determine whether NIRS can differentiate *Ae. aegypti* females infected with either ZIKV or DENV single infection, and those coinfected with ZIKV/DENV from uninfected ones. Using 200 *Ae. aegypti* females reared and infected in laboratory conditions, the training model differentiated mosquitoes into the four treatments with 100% accuracy. DENV-, ZIKV-, and ZIKV/DENV-coinfected mosquitoes that were used to validate the model could be correctly classified into their actual infection group with a predictive accuracy of 100%, 84%, and 80%, respectively. When compared with mosquitoes from the uninfected group, the three infected groups were predicted as belonging to the infected group with 100%, 97%, and 100% accuracy for DENV-infected, ZIKV-infected, and the co-infected group, respectively. Preliminary lab-based results are encouraging and indicate that NIRS should be tested in field settings to evaluate its potential role to monitor natural infection in field-caught mosquitoes.

## 1. Introduction

The transmission of arbovirus such as dengue (DENV), Zika (ZIKV), and chikungunya (CHIKV) has increased in over 50 countries in the last decade. It is estimated that 3.9 billion people from 128 countries are now at risk of DENV epidemic, whilst ZIKV and CHIKV have rapidly spread to a global public health menace [1,2,3,4]. The transmission of these arboviruses is highly associated with the distribution of their primary vector, the mosquito *Aedes aegypti*. This species dominates urban settings and is closely associated with human dwellings, laying eggs mostly in man-made breeding sites and blood-feeding preferentially in humans [5,6,7,8]. A secondary species for arboviruses transmission is *Ae. albopictus*, a species more abundant in areas with high vegetation coverage, more diverse host preferences and lower vectorial capacity than *Ae. aegypti* [9,10,11,12,13].

Arbovirus surveillance is a key element in early warning systems for epidemics as rapidly identifying and responding to outbreaks is critical for vector control strategies [14,15,16,17,18]. However, current surveillance systems most commonly rely on notifications of human infections following diagnosis by molecular assays such as Reverse Transcriptase-Polymerase Chain Reaction (RT-PCR), which can be performed on only a small fraction of suspected cases [19,20]. Rapid and cost-effective tools that can detect infections in disease vectors would be helpful to provide alert signals of arbovirus circulation before its detection in humans seeking for local health units, which lately could increase preparedness and reduce disease outbreaks among susceptible populations [21]. Traditionally, screening of natural arbovirus infection in mosquitoes is done by RT-PCR. Although accurate, RT-PCR requires skilled technicians, expensive consumables, and involves time consuming and technically demanding experiments. This limits the number of samples that can be processed to determine the level of risk to a population in a timely manner, making RT-PCR unsuitable for large-scale surveillance and early warning system for arbovirus outbreaks [18,22,23,24,25].

The Near-Infrared spectroscopy (NIRS) technique has recently been proposed as an alternative tool to improve surveillance of arbovirus vectors [26]. The NIRS technique, whose wavelength ranges from 750 to 2500 nm, measures specific frequencies of light absorbed by functional groups such as C-H, O-H, S-H, and N-H present in a biological sample. A spectrum is generated for each sample, and it is used to classify it based on the type and the concentration of these compounds. The technique is (a) non-destructive, as it preserves the biological material, (b) low cost, as it does not require reagents to operate, (c) rapid, as it can analyze a sample in just 3 s, and (d) eco-friendly, since does not produce laboratory waste.

For mosquitoes, NIRS was first applied to simultaneously predict age and cryptic species identification [26,27,28,29,30,31,32,33,34]. More recently, it has been used to screen mosquito vectors for pathogens like *Wolbachia* [35,36,37], *Plasmodium* [38,39,40], ZIKV [32], and CHIKV [37]. For instance, NIRS non-invasively detected ZIKV in laboratory reared *Ae. aegypti* with a prediction accuracy ranging from 88.8% to 99.3% [32]. NIRS can also predict age and infections in preserved [30,41,42] and fresh samples [37]. While the ability for NIRS to detect ZIKV and CHIKV has been explored, no available data reported the accuracy of predictive models for DENV, the most widespread arbovirus. Likewise, NIRS ability to detect which of two arbovirus infection is present or coinfections has not been assessed. A significant step toward evaluating the potential of NIRS to improve surveillance relies on its capacity to differentiate arbovirus infection in disease vectors. Here, we further demonstrate the accuracy of NIRS for differentiating *Ae. aegypti* females infected with either DENV or ZIKV single infection and coinfection DENV/ZIKV.

## 2. Materials and Methods

### 2.1. Mosquito Rearing

Eggs were collected through 80 ovitraps placed roughly every 25 m from each other in the Urca (22°56′43″ S; 43°09′42″ W) neighborhood, Rio de Janeiro, Brazil. Ovitraps were installed over an extensive geographic area to ensure the genetic variability of local *Ae. aegypti* was captured. The trapping period lasted four consecutive weeks. Captured mosquitoes were reared in the insectary at Laboratório de Transmissores de Hematozoários, Fiocruz, Brazil, at 27 ± 2 °C, 70 ± 5% humidity, and 12:12 h light/dark cycle. *Aedes aegypti* eggs (F2 generation, Urca, Rio de Janeiro, Brazil, lab colony) were hatched, larvae were fed on TetraMin tropical flakes (Tetra, Melle, Germany) and pupae were transferred into cages (40 cm × 40 cm × 30 cm) for adult emergence. Adults were allowed to mate for 3 to 4 days and were fed on a 10% sugar solution *ad libitum* until 36 h before conducting experimental infections.

### 2.2. Experimental Infections

Five to seven days-old, mated *Ae. aegypti* females were separated into four different groups for infection with either DENV, ZIKV, coinfection of both viruses, and negative controls. Both viral strains have a history of low passage in C6/36 cells. We used the currently circulating strain of ZIKV [BRPE243/2015 (BRPE)] in Brazil, which was isolated from a ZIKV-infected patient in late 2015 and maintained in cell culture [43], and a DENV serotype 1 strain MV17 (DENV-1) isolated from a human case at Minas Gerais State, Brazil in 2015 (DENV1/H. sapiens/Brazil Contagem-MG/MV17/2015) [44]. Viral titers were quantified via plaque-forming assay PFU (plaque forming units)/mL before experimental infection. For single infections, 1 mL of ZIKV 4.3 × 10^6^ PFU/mL or DENV 7.9 × 10^5^ PFU/mL was mixed with 1 mL of human blood (1:1). For co-infection, 1 mL of ZIKV, 1 mL of DENV (both viruses at the same titer as mentioned above) and 2 mL of human blood (1:1:2) were mixed and used to orally infect females. The control group were fed with the same blood and virus-free culture medium.

Following infection, visually engorged mosquitoes were maintained in incubators at 27 °C and 70% relative humidity. At 14-days post-infection (dpi) between 50–100 *Ae. aegypti* females from control and infected groups were randomly selected for NIR spectrum collection. Experimental infection was replicated twice to produce two distinct cohorts for the development and validation of machine learning algorithms.

### 2.3. Scanning of Mosquitoes Using NIR

Prior to scanning, mosquitoes were killed by placing them in a closed jar with an acetate-soaked cotton ball for 1 min. Whole mosquitoes were arranged on their sides on a Spectralon diffuse reflectance stage (Malvern Panalytical, Malvern, United Kingdom). The head/thorax were placed under an optic illumination fiber and scanned using a Labspec 4i NIR instrument with wavelengths ranging from 350–2500 nm (Malvern Panalytical, Malvern, United Kingdom). A reflectance spectrum was collected with an external fiber optic probe using a previously described protocol [27]. An average spectrum of 15 scans was collected for each head/thorax scanned. For each infection group and for both cohorts at least 50 mosquitoes were scanned at 14 days post incubation. The mean spectra of all treatment groups are shown in Figure 1.

### 2.4. Confirmation of ZIKV and DENV Infections by RT-qPCR

Viral RNA was extracted from individual mosquitoes randomly selected at 14 dpi, which is the estimated extrinsic incubation period for DENV in *Ae. aegypti* females [45] using a QIAamp Viral RNA Mini kit (Qiagen, Hilden, Germany). Viral RNA detection and quantification in each specimen was performed through RT-qPCR with the SuperScript III Platinum One-Step qRT-PCR Kit (Invitrogen, Waltham, MA, USA) in QuantStudio 6 Flex Real-Time PCR System (Applied Biosystems, Waltham, MA, USA) using previously published primers and amplification conditions [23,32]. At least 3–4 uninfected mosquitoes from the lab controls were added as negative controls in every 96-well plate. Virus copy numbers were calculated through absolute quantification in each run using a standard curve of a seven-point dilution series (10^2^ to 10^8^ copies) of in vitro transcribed ZIKV or DENV RNA [43]. Such an approach allowed us to correlate the accuracy of NIRS with a range of DENV and ZIKV titers.

### 2.5. Data Analysis

Raw spectra were organized in excel and exported into JMP (Version 16 software, SAS Institute Inc., Cary, NC, USA) for analysis. To assess the capacity of NIRS to differentiate DENV, ZIKV single infection, and ZIKV/DENV coinfected from uninfected *Ae. aegypti*, mosquito samples were divided into a training set comprising 60% of the data set (*n* = 359) and a test set. The training set was further split into training (75%) and validation sets, i.e., samples seen by the model but not used to train the model (25% of the data). The remaining data (*n* = 159) was used as a test set i.e., samples not seen by the model to validate the accuracy of the model.

Predictive analytics were developed using artificial neural networks (ANN). These networks were fully connected, consisting of an input layer, an output layer and 1 hidden layer and a TanH activation function. A single model was generated to differentiate mosquitoes in the 4 groups.

The number of viral copies in mosquito samples was not normally distributed (Shapiro-Wilk *W* = 0.28, *p* < 0.001) and thus virus quantities observed for the different conditions (ZIKV and DENV copies in mono infections, and in co-infection) were compared with Kruskal–Wallis rank sum test (H), followed by pairwise comparisons using Wilcoxon rank sum test (U-test) in the R environment [46].

### 2.6. Ethical Considerations

*Aedes aegypti* colonies were maintained in the lab by blood-feeding of anonymous donors acquired from the Rio de Janeiro State University blood bank. The blood bags were rejected from the bank due to small blood volume. No information on the donors (including sex, age, and clinical condition) was disclosed. The use of human blood was approved by the Fiocruz Ethical Committee (CAAE 53419815.9.0000.5248).

## 3. Results

### 3.1. Infection Results

A subset of 96 mosquitoes for either ZIKV-infected and DENV-infected groups and 88 insects for ZIKV/DENV coinfected were individually screened for both viruses at 14 dpi with RT-qPCR. The average infection rate was ~90% for all groups: 87, 87, and 85 *Ae. aegypti* females were confirmed positive by RT-PCR for ZIKV and DENV single infection and for ZIKV/DENV coinfection, respectively (Table 1). Remarkably, the number of DENV copies were similar between single infection and ZIKV/DENV coinfection (U-test, *p* = 0.35), whereas a statistically significant higher number of ZIKV copies were observed in co-infection than in single ZIKV infection (H = 162, df = 3, *p* < 0.001; U-test, *p* < 0.001). In ZIKV/DENV co-infected mosquitoes, the number of ZIKV copies was higher than the number of DENV copies at 14 dpi (U-test, *p* < 0.001) (Figure 2).

### 3.2. Differentiation between DENV, ZIKV, and ZIKV + DENV Coinfections in Ae. aegypti

The training model was built using a total of 200 *Ae. aegypti* females, with between 50–60 individuals belonging to each of the four treatments we used: DENV-infected, ZIKV-infected, ZIKV/DENV coinfected and uninfected mosquitoes. The training model was able to differentiate *Ae. aegypti* mosquitoes into the four treatments of samples with 100% accuracy. In the training model the accuracy for correctly identifying the uninfected individuals was of 98% (*n* = 50) (Table 2).

When the model was applied to predict the test set, DENV-infected, ZIKV-infected, and ZIKV/DENV coinfected mosquitoes could be correctly classified into their actual infection group with a predictive accuracy of 100% (*n* = 37), 84% (*n* = 37) and 80% (*n* = 35), respectively. Furthermore, when compared with insects from the uninfected group, the three infected groups were predicted as belonging into the infected group with 100%, 97%, and 100% accuracy for DENV-infected, ZIKV-infected and the coinfected group, respectively. Uninfected mosquitoes were predicted with 80% accuracy (*n* = 50).

In summary, NIRS was highly sensitivity for detecting any infection type and could distinguish ZIKV and DENV infected mosquitoes from uninfected samples and samples coinfected with both viruses. Prediction accuracy for the training set and the validation set is shown in Figure 3 and Table 2, whereas test set data is shown in Figure 4 and Table 2.

### 3.3. Second Derivative Figure of Infected and Uninfected Mosquitoes

Figure 5 shows the second derivative average spectra of all ZIKV-infected, DENV-infected, ZIK-DENV- coinfected, and uninfected mosquitoes with distinct bands from the first overtone region at 1685 and 1741 and the combination band region at 1868, 1890, 2234, 2256, 2301, and 2320 nm. Bands from the first overtone region are dominated by C-H and O-H, while bands within the combination region are dominated by C-H, N-H, and O-H bonds representing proteins. A general change in absorbance is observed from the raw spectra as well as the second derivative spectra. Reduced bands among infected mosquitoes in particular those infected with ZIKV are observed at 1685, 2256, and 2301 nm whereas bands at 1741 and 1868 nm are increased in uninfected mosquitoes. Bands at 2234 and 2320 nm are increased in ZIKV infected mosquitos relative to other mosquitoes.

## 4. Discussion

This study sought to test the capacity of the NIRS technique to differentiate *Ae. aegypti* female mosquitoes single infected with either ZIKV or DENV, coinfected with both ZIKV and DENV and uninfected samples. NIRS has previously been used to detect single infections of ZIKV [32], CHIKV [37], and DENV [47] in laboratory-reared *Ae. aegypti* mosquitoes. Here, our results show that NIRS can also differentiate DENV-infected *Ae. aegypti* females from their uninfected counterparts, with an accuracy of 100%. Additionally, this manuscript also shows that NIRS can detect and differentiate DENV and ZIKV infected mosquitoes from ZIKV/DENV coinfections and uninfected *Ae. aegypti* with high predictive sensitivities > 96%. Preliminary results obtained so far have shown that predictive models using *Ae. aegypti* mosquitoes experimentally infected under laboratory conditions have accuracies above 92% in detecting DENV, ZIKV, and CHIKV, three of the most widespread arboviruses. Taken together, our results are encouraging to continue evaluating the potential role of NIR to improve arboviruses surveillance in endemic settings. Thus, next steps should involve collection of semi-field or field data to move further with estimating the potential role that NIRS might have in arboviruses surveillance systems [17,18].

As far as we know, the results presented here are the first to show that NIR can detect and differentiate *Ae. aegypti* infected with single ZIKV or DENV infection and ZIKV/DENV co-infection of uninfected insects. The use of spectroscopy techniques at the infrared light for arboviruses diagnosis is recent. The overall high predictive ability observed for DENV are consistent with previous findings where ZIKV infected *Ae. aegypti* could be detected in fresh samples with a sensitivity > 95% in head and thoraces [32] and in samples left in BG sentinel traps for 7 days with sensitivity of 93.2% [37]. These results are encouraging to applying this technique in further tests under more realistic scenarios like semi-field or field testing. Actual evidence shows NIRS might become a relevant tool for surveillance of arbovirus programs in areas where both ZIKV and DENV occur simultaneously like in several tropical regions worldwide [48,49,50].

The occurrence of natural coinfection of arboviruses in *Ae. aegypti* is still poorly known, although several studies have demonstrated both in the laboratory and under field settings that mosquitoes are able to be simultaneously infected by two different arboviruses [51,52,53,54,55,56]. It has also been reported that the presence of a coinfection does not affect *Ae. aegypti* vector competence for each pathogen [53,54], but the presence of one virus can enhance the replication of the other virus in the salivary gland. For example, a co-infection of DENV and CHIKV enhanced the replication of DENV in the salivary glands [55]. Our data shows that the number of ZIKV copies were increased in *Ae. aegypti* when insects were co-infected with DENV. Ruckert and colleagues also demonstrated that both ZIKV infection rates and DENV dissemination rates were reduced for *Ae. aegypti* with a coinfection of CHIKV whereas CHIKV transmission was reduced with DENV infection [54]. The sensitivity for ZIKV infected mosquitoes was 84% which was lower than the sensitivity for DENV prediction (100%). While the sensitivity of ZIKV was lower, samples incorrectly predicted as ZIKV positive were predicted as infected with DENV instead, suggesting infection was still identified, albeit the incorrect one. As we only collected NIR spectra from the heads and thoraces of mosquitoes at 14 dpi, it is possible that the presence of DENV particularly in the salivary glands might be higher than ZIKV, even though the overall quantity of ZIKV from mosquito whole body was higher than DENV as determined by RT-PCR (Figure 2). We hypothesized that these factors may have resulted in a more accurate training model for DENV-infected mosquitoes, which eventually misclassified ZIKV-infected mosquitoes as DENV-infected. ZIKV and DENV are closely related viruses within the Flaviviridae family sharing 55.6% amino acid sequence identity [57], with 79 genes that were regulated in the same direction (up or downregulation) during *Ae. aegypti* single infections with both viruses [58]. Such similarities between DENV and ZIKV are also evident from the overlapping raw spectra of single-infected *Ae. aegypti* mosquitoes (Figure 1). Regardless of the infection type, NIR was 84% sensitive for predicting presence of infection in ZIKV infected mosquitoes. This sensitivity is consistent with what was previously reported by the last two studies [32,37]. Another possible explanation for the misclassification of ZIKV into DENV-infected mosquitoes might be related to the time in dpi the qPCR screening was conducted. The duration of the extrinsic incubation period (EIP) varies between these viruses. The EIP for DENV is estimated to last 14 days [45], whereas the EIP for ZIKV is shorter and estimated as approximately 10 days [59]. Therefore, by selecting to scan mosquitoes at 14 dpi, we could unintentionally favor a misclassification toward DENV because a peak on its viral load at mosquito body is expected for 14 dpi, whereas the shorter EIP for ZIKV could make that on 14 dpi, its viral load might not be on its peak anymore [45,59].

The NIR measures specific frequencies of light absorbed between 750–2500 nm and is capable to differentiate infected from uninfected disease vectors most likely by assessing the physiological and biochemical changes due to pathogen presence. For example, the detection of the endosymbiont *Wolbachia pipientis* in both *D. simulans* and *D. melanogaster* has been attributed to spectral signatures related to either the presence and the concentration of lipopolysaccharide molecules or the physiological changes caused by this bacterium in infected flies [60]. In regards of ZIKV and DENV infection in *Ae. aegypti*, physiological changes due to arbovirus infection have been reported using different approaches. *Aedes aegypti* females when infected with DENV or ZIKV have a reduction in life-history traits such as survivorship, fecundity, and fertility [61,62,63,64,65]. On a molecular and biochemical level, there are abundant data using gene expression, transcriptomics, and proteomics approaches revealing that most of the altered genes are involved in metabolic processes, cellular processes, and proteolysis [58,66,67,68]. Thus, most likely the NIR detects those physiological changes caused by infection and then is able to differentiate between infected and uninfected individuals.

NIRS represents an ideal arbovirus surveillance tool as it can be implemented for real time screening of field mosquitoes, with remarkable advantages such as cost and time effectiveness, reagent-free, and non-invasive nature over existing molecular tools for similar purposes. Previously, it was demonstrated that NIRS is 18 times faster and 110 times cheaper than RT-qPCR for detecting ZIKV in *Ae. aegypti* mosquitoes [32]. Its high throughput nature allows hundreds of samples to be screened in a day by unskilled technicians ultimately enabling timely decision making through rapid detection of potential disease. This is particularly important when considering the silent circulation of arboviruses before an outbreak within the human population is detected, leading to the possibility of early detection of the virus within the mosquito [69,70,71,72].

Future work should focus on establishing identity of the chemical signatures preferably by scanning pure viruses. Identification of virus specific spectral peaks will speed up the implementation of the tool for field collected mosquitoes. This study and the previous two studies that used NIRS to predict ZIKV in fresh *Ae aegypti* [32] and ZIKV, *Wolbachia,* and CHIKV in mosquitoes left in BG sentinel trap for 7 days [37] provide strong ground to progress the application of NIRS to assist with the assessments of arbovirus transmission studies that require processing of hundreds of samples in the laboratory. However, the use of NIRS in the field to detect mosquitoes with arboviruses will require further validation including training of fresh models built directly from field mosquitoes infected with these viruses.

## Figures and Tables

**Figure 1 viruses-15-00011-f001:**
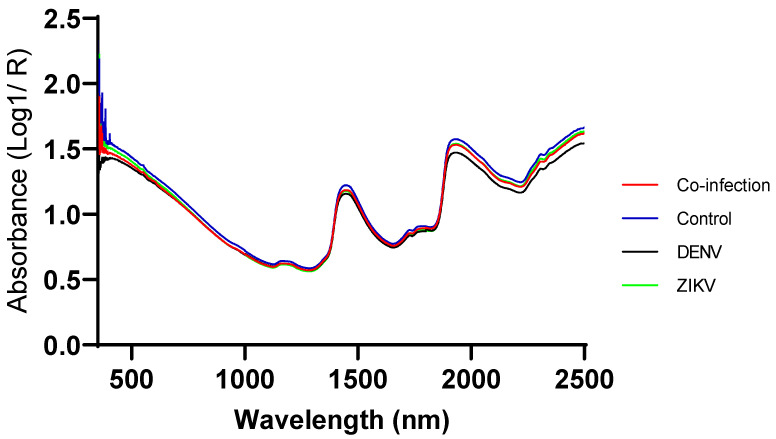
Average NIR spectra of Co-infected, ZIKV, DENV and uninfected mosquitoes (control) collected at 14 days post-infection by the Labspec 4i NIR spectrometer. In particular the absorbance value of uninfected mosquitoes (Blue line) was slightly higher than those of infected mosquitoes.

**Figure 2 viruses-15-00011-f002:**
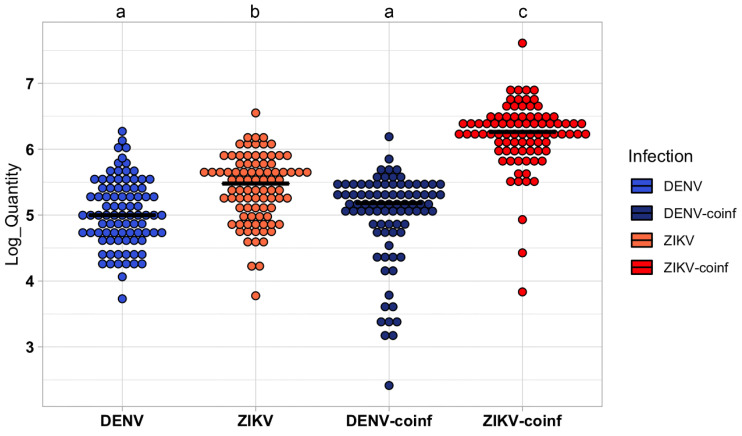
The number of viral copies in *Aedes aegypti* females at 14 dpi when mosquitoes were either single infected with ZIKV or DENV, or simultaneously co-infected with both ZIKV/DENV. Different letters indicate statistically significant differences (Wilcoxon rank sum test (U-test), *p* < 0.001). Each circle represents an individual mosquito; negative control samples (uninfected *Ae. aegypti* from laboratory colony) are not shown because they did not produce any amplification signal.

**Figure 3 viruses-15-00011-f003:**
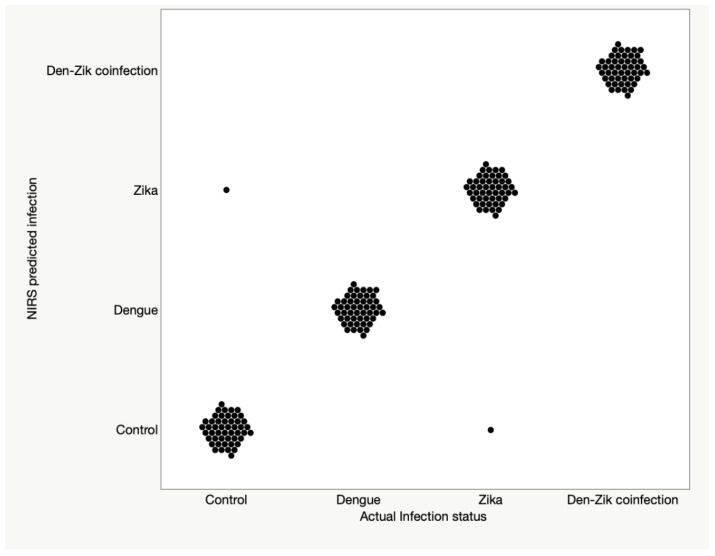
The predictive accuracy of NIRS for samples that were used for training and validation of the ANN model.

**Figure 4 viruses-15-00011-f004:**
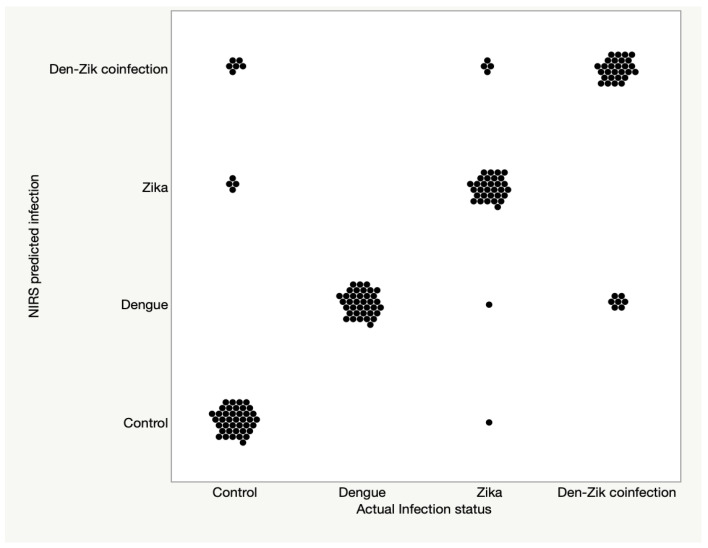
The predictive accuracy of NIRS for samples that were used to test the ANN model.

**Figure 5 viruses-15-00011-f005:**
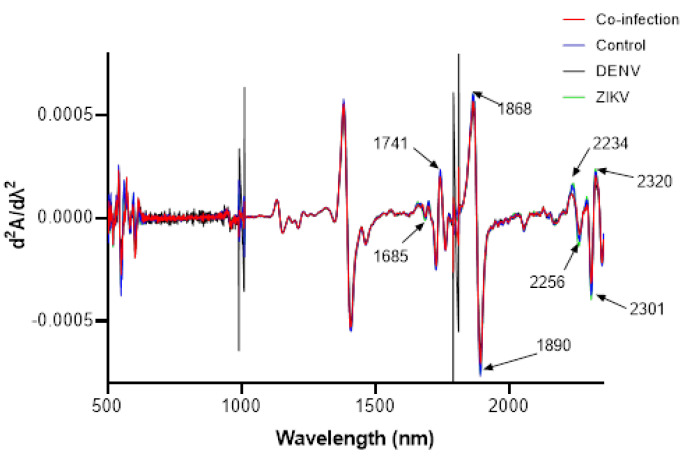
The second derivative spectra of *Aedes aegypti* mosquitoes showing overtones of infected and uninfected groups. The black arrows in the figure indicate the wavelength (nm) band regions that are important for differentiating the groups.

**Table 1 viruses-15-00011-t001:** Infection rate for DENV or ZIKV single infection and DENV/ZIKV coinfection.

	Group		
	ZIKV-Infected	DENV-Infected	ZIKV/DENV Coinfected
Infection rate	90.62% (*n* = 87/96)	90.62% (*n* = 87/96)	96.6% (*n* = 85/88)

**Table 2 viruses-15-00011-t002:** Prediction accuracy for *Ae. aegypti* infection status in training/validation set and test set. Mosquitoes could be infected with either DENV or ZIKV single infection, coinfected with DENV and ZIKV or uninfected (control).

	Training and Validation Set				Test Set			
	% Spec (*n)*	% Sensitivity (*n)*			% Spec (*n)*	% Sensitivity (*n)*		
Predicted Group	Control	DENV	ZIKV	Co-Infection ZIKV/DENV	Control	DENV	ZIKV	Co-Infection ZIKV/DENV
Predicted into actual infection group	98 (*n* = 50)	100 (*n* = 50)	98 (*n* = 50)	100 (*n* = 50)	80 (*n* = 50)	100 (*n* = 37)	84 (*n* = 37)	80 (*n* = 35)
Predicted as infected	2 (*n* = 50)	100 (*n* = 50)	98 (*n* = 50)	100 (*n* = 50)	20 (*n* = 50)	100 (*n* = 37)	97 (*n* = 37)	100 (*n* = 35)

## Data Availability

Not applicable.

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
