# Peer review of "Rapid and Non-Invasive Detection of Aedes aegypti Co-Infected with Zika and Dengue Viruses Using Near Infrared Spectroscopy"

_viruses, 2022, doi:10.3390/v15010011_

Round 1

Reviewer 1 Report

In this manuscript, the authors describe Rapid and non-invasive detection of Aedes aegypti co-infected with Zika and dengue viruses using Near Infrared Spectroscopy. It is generally well written, and the preliminary lab-based results are interesting and provide an opportunity and basis for further more rigorous testing in field settings.

Major points:

·      Why was DENV-1 chosen? Would be interesting to see the NIR spectra for other serotypes. Do the authors expect other serotypes to show different results?

·      Line 122 - For each infection group and for both cohorts at least 50 mosquitoes were scanned at 7, 10 and/or 14 days post incubation.

o   The and/or is confusing? When were the scans done? Different days for different groups, cohorts?

o   Should be dpi

·      Figure 1 should mention the day (7, 10, 14 or average of those?), the NIR spectra were scanned.

·      Line 167-168 – the numbers of mosquitoes said to be screened does not match with the N numbers mentioned in table 1.

·      Figure 4 – how do the authors explain the NIRS predicted infection of DENV-ZIKV coinfection for the Control actual infection status?

·      The authors should rephrase the following sentences to make their point clear.

o   Line 271 - This would have resulted into a training model that was more accurate for DENV infected mosquitoes which misclassified ZIKV infected mosquitoes as DENV infected.

o   Line 309  –This is particularly important when we consider the undetected circulation of arboviruses in mosquito populations before an outbreak within the human population is detected.

·      How would the spectra look like when comparing infected Aedes aegypti vs Aedes albopictus? Or when comparing infected mosquitos of different ages? Would spectra differences observed in these lab controlled experiments still be significant?

Minor points (suggested changes are in bold):

·      Aedes aegypti or Ae. Aegypti – the authors should choose one way to write it.

·      Line 104 - DENV titer depiction as DENV 7.9x105.25 PFU/ml seems oddly specific. Why not 106?

·      Line 110 – extra space in 27 oC

·      In materials and methods the wavelengths for NIR instrument is stated to range from 350-2500nm. In discussion it is mentioned to be between 750-2500nm.

·      Line 145 - … mosquito samples were divided into a training set, comprising 60% of the data set (n=359), and a test set.

·      Line 149 – missing period after model.

·      Line 160 - Aedes aegypti colonies were maintained in the lab…

·      Figure 2 is not referenced in the text. Should be mentioned in section 3.1.

·      Line 180 - …when mosquitoes were either…

·      DENV, ZIKV, CHIKV and dengue, Zika, chikungunya are used interchangeably in the text, figures, and tables. Should be consistent throughout the manuscript.

·      Line 242 – the sentence should be rephrased for clarity.

·      Line 255 - … field settings that mosquitoes are able to…

·      Line 260 – what do the authors mean by opposite pattern? The previous sentence states an enhancement, and their data shows an increase.

·      Line 271 – extra space before figure 2.

·      Line 275 -what do the authors mean by…genes that were regulated in the same direction…?

·      Line 281 - …related to the time in dpi the qPCR..

·      Line 297 - … there are abundant data using gene expression, transcriptomics and proteomics approaches revealing that…

·      Line 318 – extra space before However

Author Response

In the behalf of all authors, I would like to appreciate Reviewer effort in improving the quality of our manuscript. All suggestions were accepted, as described below.

Reviewer #1:

In this manuscript, the authors describe Rapid and non-invasive detection of Aedes aegypti co-infected with Zika and dengue viruses using Near Infrared Spectroscopy. It is generally well written, and the preliminary lab-based results are interesting and provide an opportunity and basis for further more rigorous testing in field settings.

Major points:

  • Why was DENV-1 chosen? Would be interesting to see the NIR spectra for other serotypes. Do the authors expect other serotypes to show different results?

We used DENV-1 based on previous data from our group showing this specific dengue sample shows high infection rates of laboratory-reared Ae. aegypti. We believe the difference between ZIKV, DENV and coinfected ZIKV/DENV using NIR technique should remain similar using other dengue serotypes besides DENV-1. Indeed, further investigations are needed to better mensurate the efficacy of NIRS in different conditions and scenarios. Studies comparing the spectra of different DENV serotypes using NIR are ongoing in our laboratory and the first data are expected soon.

  • Line 122 - For each infection group and for both cohorts at least 50 mosquitoes were scanned at 7, 10 and/or 14 days post incubation.

o   The and/or is confusing? When were the scans done? Different days for different groups, cohorts?

The scans were done only at 14 days post incubation period for all analyzed groups. The correction information was added in the manuscript.

  • Should be dpi

Done as recommended.

  • Figure 1 should mention the day (7, 10, 14 or average of those?), the NIR spectra were scanned.

We used the spectra for 14 dpi only. The correct information was added in Figure 1 legend and throughout the manuscript.

  • Line 167-168 – the numbers of mosquitoes said to be screened does not match with the N numbers mentioned in table 1.

We added in Table 1 the total number of screened mosquitoes and also the number of confirmed infected mosquitoes by PCR, to make it clear.

  • Figure 4 – how do the authors explain the NIRS predicted infection of DENV-ZIKV coinfection for the Control actual infection status?

Once the uninfected status for the Control group was confirmed by PCR, we believe that the technique sometimes misclassified samples. Although uncommon, sometimes false positive results are obtained, showing a decrease in specificity.

  • The authors should rephrase the following sentences to make their point clear.

o   Line 271 - This would have resulted into a training model that was more accurate for DENV infected mosquitoes which misclassified ZIKV infected mosquitoes as DENV infected.

We replaced to: 'We hypothesized that these factors may have resulted in a more accurate training model for DENV-infected mosquitoes, which eventually misclassified ZIKV-infected mosquitoes as DENV-infected.' (line 326)

o   Line 309  –This is particularly important when we consider the undetected circulation of arboviruses in mosquito populations before an outbreak within the human population is detected.

This is particularly important when considering the silent circulation of arboviruses before an outbreak within the human population is detected, leading to the possibility of early detection of the virus within the mosquito populations (line 374).

  •  

 How would the spectra look like when comparing infected Aedes aegypti vs Aedes albopictus? Or when comparing infected mosquitos of different ages? Would spectra differences observed in these lab controlled experiments still be significant?

            These are interesting points. So far, only two studies using NIRS have been conducted with Aedes albopictus. None of them involve mosquito infection but age grade prediction. Regarding aging, our own group showed NIR successfully separates infected from uninfected individuals with high accuracies for Aedes aegypti infected with ZIKV at 4, 7 and 10 dpi (Fernandes et al. 2018) and with CHIKV on 1-7 days after mosquito death (Santos et al. 2021). The effects of age on DENV detection will be further investigated by our group.

Minor points (suggested changes are in bold):

  • Aedes aegyptior Ae. Aegypti – the authors should choose one way to write it.

For aesthetic reasons, we only used Aedes aegypti when we initiate a sentence or in tables. The rest is Ae. aegypti.

  • Line 104 - DENV titer depiction as DENV 7.9x105.25 PFU/ml seems oddly specific. Why not 106?

We have simplified the value to 105.

  • Line 110 – extra space in 27 oC

Done as recommended.

  • In materials and methods the wavelengths for NIR instrument is stated to range from 350-2500nm. In discussion it is mentioned to be between 750-2500nm.

The equipment used in this study (Labspec 4i NIR instrument) measures the wavelengths ranging from 350-2500nm. Since the near-infrared region starts at ~750 nm, our equipment also evaluates some information from the visible region. In our analysis we can cut the spectrum and use the wavelength of interest. In this study, the region used was from 500 to 2500 nm, as shown in Figure 5.

  • Line 145 - … mosquito samples were divided into a training set, comprising 60% of the data set (n=359), and a test set.

Done as recommended.

  • Line 149 – missing period after model.

Done as recommended.

  • Line 160 - Aedes aegypti colonies weremaintained in the lab…

Done as recommended.

  • Figure 2 is not referenced in the text. Should be mentioned in section 3.1.

Done as recommended.

  • Line 180 - …when mosquitoes wereeither…

Done as recommended.

  • DENV, ZIKV, CHIKV and dengue, Zika, chikungunya are used interchangeably in the text, figures, and tables. Should be consistent throughout the manuscript.

Done as recommended.

  • Line 242 – the sentence should be rephrased for clarity.

As far as we know, the results presented here are the first to show that NIR can detect and differentiate Ae. aegypti infected with single ZIKV or DENV infection and ZIKV/DENV co-infection of uninfected insects.

  • Line 255 - … field settings thatmosquitoes are able to…

Done as recommended.

  • Line 260 – what do the authors mean by opposite pattern? The previous sentence states an enhancement, and their data shows an increase.

We rephrased to: “Our data shows that the number of ZIKV copies were increased in Ae. aegypti when insects were co-infected with DENV” (line 321).

  • Line 271 – extra space before figure 2.

Done as recommended.

  • Line 275 -what do the authors mean by…genes that were regulated in the same direction…?

We added the term up or downregulated to the sentence.

  • Line 281 - …related tothe time in dpi the qPCR..

Done as recommended.

  • Line 297 - … there are abundant datausing gene expression, transcriptomics and proteomics approaches revealing that…

Done as recommended.

  • Line 318 – extra space before However

Reviewer 2 Report

This work explores the feasibility of using NIRS to assess dengue, Zika and dengue/Zika co-infections in Ae. aegypti mosquitoes. This is the continuation of an extensive body of work by this research team exploring the ability to classify a range of physiological states in the mosquito. The authors verified that both dengue and Zika infected mosquitoes can be differentiated from uninfected mosquitoes using NIRS, both of which have been previously described. They also demonstrated that they could successfully identify mosquitoes co-infected with both dengue and Zika. Technically this manuscript is fairly sound, although I have some minor points detailed below. However, my greatest concern is whether this work will ever be able to identify infected mosquitoes under field conditions. This has been an ongoing issue with NIRS categorization. In the highly controlled and homogeneous environment of the lab NIRS is quite capable of determining age, infection status, etc. However, when trying to categorize mosquitoes in the field with unknown ages, nutritional status, reproductive status, immune challenges, etc., will NIRS still be able to overcome all of this inevitable variation to focus on the single factor of arboviral infection status. I think it is now incumbent on the authors to turn the corner and determine the feasibility of this approach at least under semi-filed conditions. For example, the authors did use field collected mosquito eggs, which is a great step, but their adult age was homogeneous and we know that the NIRS spectra changes as mosquitoes age. Will this mask the identification of the virus in infected mosquitoes?

Minor concerns:

The authors diluted the blood by 50% with culture medium. How does this significant dilution impact nutritional status and fecundity?

Were mosquitoes tested for dissemination of the virus from the midguts and to the salivary glands?

Generating NIRS models with only 50 mosquito spectra seems quite low. With that said it certainly appeared to work under lab conditions so it is acceptable. However, moving forward additional mosquitoes may need to be modeled to capture a greater degree of variation.

In the discussion the authors suggest that by scanning the head/thorax they are selecting for mosquitoes with virus in the salivary glands (e.g., Line 284: Therefore, by selecting to sample mosquitoes at 14 dpi, we could unintentionally favor higher viral load at the salivary glands of mosquitoes, the tissue/organ we scanned for NIR”. You did not scan the salivary glands, the cuticle of the head and thorax was scanned. Changes to the cuticles due to the infection may be detected, but that could be due to any stage of arboviral infection (eg. midgut infection). This is actually described by them at the end of the next paragraph where they state “Thus, most likely the NIR detects those physiological changes caused by infection and then is able to differentiate between infected and uninfected individuals”. This needs to be corrected.

The recently published article: “Santos, Marfran CD, et al. "Infrared spectroscopy (NIRS and ATR-FTIR) together with multivariate classification for non-destructive differentiation between female mosquitoes of Aedes aegypti recently infected with dengue vs. uninfected females." Acta Tropica 235 (2022): 106633.” Is directly relevant to this work and should be included as a reference.

In summary this research is sound, but the authors need to move beyond repeated lab experiments and verify whether these effects can be detected in field populations. In its current for this manuscript only expands our knowledge of arboviral identification with NIRS by demonstrating that mosquitoes co-infected with Zika and dengue can be differentiated. This is a fairly minor addition and thus limits my enthusiasm for this work.

Author Response

On the behalf of the other authors, I would like to appreciate the efforts of Reviewer in improving the quality of our manuscript. All suggestions were accepted, as you can see below.

Reviewer #2:

This work explores the feasibility of using NIRS to assess dengue, Zika and dengue/Zika co-infections in Ae. aegypti mosquitoes. This is the continuation of an extensive body of work by this research team exploring the ability to classify a range of physiological states in the mosquito. The authors verified that both dengue and Zika infected mosquitoes can be differentiated from uninfected mosquitoes using NIRS, both of which have been previously described. They also demonstrated that they could successfully identify mosquitoes co-infected with both dengue and Zika. Technically this manuscript is fairly sound, although I have some minor points detailed below. However, my greatest concern is whether this work will ever be able to identify infected mosquitoes under field conditions. This has been an ongoing issue with NIRS categorization. In the highly controlled and homogeneous environment of the lab NIRS is quite capable of determining age, infection status, etc. However, when trying to categorize mosquitoes in the field with unknown ages, nutritional status, reproductive status, immune challenges, etc., will NIRS still be able to overcome all of this inevitable variation to focus on the single factor of arboviral infection status. I think it is now incumbent on the authors to turn the corner and determine the feasibility of this approach at least under semi-filed conditions. For example, the authors did use field collected mosquito eggs, which is a great step, but their adult age was homogeneous and we know that the NIRS spectra changes as mosquitoes age. Will this mask the identification of the virus in infected mosquitoes?

We appreciate Reviewer #2 comments. We agree that it is necessary to give a step further and demonstrate the effectiveness of the technique in a field or semi-field setting. However, we believe that this work presents a great advance in studies of NIR technique application for arbovirus surveillance. A common metaphor for the accumulation of scientific knowledge is of research projects as the bricks from which a wall is built. Each study contributes to the growing structure as “another brick in the wall”. In our case it is not different. The NIR has been used in areas such as astronomy, food science, arts, chemistry, etc, but only recently on vector-borne diseases. The paper published in 2018 by Fernandes et al. (2018) in Science Advances showed for the first time the ability of NIRS in separating ZIKV-infected from uninfected Ae. aegypti. The finding that NIR is able to differentiate mosquitoes when infected with either ZIKV or DENV single infection, and simultaneously coinfected with ZIKV and DENV from uninfected individuals is an additional brick in that wall. We hope that after a few years, the wall could answer if NIR can be used as a surveillance tool. But to get there, we will not go at once, but step by step. Additionally, as any other approach used to mitigate disease transmission, research starts in a controlled environment and only if successful in those restricted conditions, the next step is taken. We need to develop prediction models in controlled scenarios before moving forward to the field. If we fail in this first step, there is no need to move further. Likewise, insecticides, Wolbachia, and gene-driving systems, for instance, have been subjected to great scrutiny before going to the field. We are aware of the limitations of our results and acknowledged it on the Discussion section, including the challenge of going to a semi-field or field scenario to test whether lab-based models are able to effectively detect infection in field-caught mosquitoes.

Minor concerns:

The authors diluted the blood by 50% with culture medium. How does this significant dilution impact nutritional status and fecundity?

We appreciate the observation of Reviewer, but considering arboviruses like DENV and ZIKV only grows in culture medium there is no other way of allowing the viral particles to infect red blood cells than mixing the medium and the blood. This proportion is part of experimental infection of several research groups able to infect mosquitoes with arboviruses. I.e., if we don't add the cell culture in a blood sample, we can only infect mosquitoes with an intrathoracic injection. The controls are orally fed with blood + cell culture in the absence of virus to keep the protocol and allow further comparisons among the spectra collected in all situations.

 Were mosquitoes tested for dissemination of the virus from the midguts and to the salivary glands?

In this study the arbovirus dissemination throughout Ae. aegypti was not evaluated, only the whole body of mosquitoes was analyzed by RT-qPCR. This choice was made because other works done by our group showed high susceptibility to both DENV-1 and ZIKV strains used in this paper, with the same mosquito population from Rio de Janeiro (Silveira et al. 2018). Later, our results confirmed the high susceptibility of local Ae. aegypti to the two viruses used.

Generating NIRS models with only 50 mosquito spectra seems quite low. With that said it certainly appeared to work under lab conditions so it is acceptable. However, moving forward additional mosquitoes may need to be modeled to capture a greater degree of variation.

We agree with the comment made. We are aware that it will be necessary to increase the number of spectra analysed when the technique move forward the field studies to encompass a greater degree of variation. This point will be considered when moving on to the next steps in semi-field/field conditions.

 In the discussion the authors suggest that by scanning the head/thorax they are selecting for mosquitoes with virus in the salivary glands (e.g., Line 284: Therefore, by selecting to sample mosquitoes at 14 dpi, we could unintentionally favor higher viral load at the salivary glands of mosquitoes, the tissue/organ we scanned for NIR”. You did not scan the salivary glands, the cuticle of the head and thorax was scanned. Changes to the cuticles due to the infection may be detected, but that could be due to any stage of arboviral infection (eg. midgut infection). This is actually described by them at the end of the next paragraph where they state “Thus, most likely the NIR detects those physiological changes caused by infection and then is able to differentiate between infected and uninfected individuals”. This needs to be corrected.

We appreciate Reviewer observation. We rephrased the sentence to be clearer: “Therefore, by selecting to scan mosquitoes at 14 dpi, we could unintentionally favor a misclassification toward DENV because a peak on its viral load at mosquito body is expected for 14 dpi, whereas the shorter EIP for ZIKV could make that on 14 dpi, its viral load might not be on its peak anymore.” (line 346).

 The recently published article: “Santos, Marfran CD, et al. "Infrared spectroscopy (NIRS and ATR-FTIR) together with multivariate classification for non-destructive differentiation between female mosquitoes of Aedes aegypti recently infected with dengue vs. uninfected females." Acta Tropica 235 (2022): 106633.” Is directly relevant to this work and should be included as a reference.

Yes, we agree. This study came out after our submission. The work has been included in the discussion now.

 In summary this research is sound, but the authors need to move beyond repeated lab experiments and verify whether these effects can be detected in field populations. In its current for this manuscript only expands our knowledge of arboviral identification with NIRS by demonstrating that mosquitoes co-infected with Zika and dengue can be differentiated. This is a fairly minor addition and thus limits my enthusiasm for this work.

 We are aware of the limitations of our results and acknowledged it on the Discussion section, including the challenge of going to a semi-field or field scenario to test whether lab-based models are able to effectively detect infection in field-caught mosquitoes. However, the finding that NIR is able to differentiate mosquitoes when infected with either ZIKV or DENV single infection, and simultaneously coinfected with ZIKV and DENV from uninfected, is an additional crucial information for move forward to field studies, as mentioned before in this review.